# l-Lysine-Based Gelators for the Formation of Oleogels in Four Vegetable Oils

**DOI:** 10.3390/molecules27041369

**Published:** 2022-02-17

**Authors:** Qiannan Li, Jieying Zhang, Guiju Zhang, Baocai Xu

**Affiliations:** School of Light Industry, Beijing Technology and Business University, No. 11 Fucheng Road, Haidian District, Beijing 100048, China; liqiannan0611@163.com (Q.L.); 13716454528@163.com (J.Z.)

**Keywords:** N^α^, N^ε^-diacyl-l-lysine, vegetable oil, oleogel, gelator, oil-binding capacity

## Abstract

Supramolecular oleogel is a soft material with a three-dimensional structure, formed by the self-assembly of low-molecular-weight gelators in oils; it shows broad application prospects in the food industry, environmental protection, medicine, and other fields. Among all the gelators reported, amino-acid-based compounds have been widely used to form organogels and hydrogels because of their biocompatibility, biodegradation, and non-toxicity. In this study, four N^α^, N^ε^-diacyl-l-lysine gelators (i.e., N^α^, N^ε^-dioctanoyl-l-lysine; N^α^, N^ε^-didecanoyl-l-lysine; N^α^, N^ε^-dilauroyl-l-lysine; and N^α^, N^ε^-dimyristoyl-l-lysine) were synthesized and applied to prepare oleogels in four kinds of vegetable oils. Gelation ability is affected not only by the structure of the gelators but also by the composition of the oils. The minimum gel concentration (MGC) increased with the increase in the acyl carbon-chain length of the gelators. The strongest gelation ability was displayed in olive oil for the same gelator. Rheological properties showed that the mechanical strength and thermal stability of the oleogels varied with the carbon-chain length of the gelators and the type of vegetable oil. The microstructure of oleogels is closely related to the carbon-chain length of gelators, regardless of oil type. The highest oil-binding capacity (OBC) was obtained in soybean oil for all four gelators, and N^α^, N^ε^-dimyristoyl-l-lysine showed the best performance for entrapping oils.

## 1. Introduction

Gel is a solid-like form of soft matter comprised of a liquid phase as the main component and a low concentration of a molecular gelator [1]. When the mass of the gelator is less than 2000 Da, the corresponding gel is defined as supramolecular gel or low-molecular-weight gel (LMWG) [2,3,4,5,6], which is formed by small molecular gelators through supramolecular interactions, including hydrogen-bonding, van der Waals, π-stacking, coordination, donor-acceptor, and charge-transfer interactions.

Two types of gels are commonly described in the literature: organogels and hydrogels [7]. Recently, ionic liquid gels, i.e., gels obtained in ionic liquids [8,9,10] and eutectogels, i.e., gels obtained in deep eutectic solvents [11,12] have also been described. Organogels are gels in which the liquid phase is an organic solvent, unlike hydrogels, which are gels with a continuous liquid-aqueous phase. Supramolecular organogels can be used in various everyday applications, including processing petroleum, food, pharmaceuticals, or cosmetics [13,14,15]. If liquid oils are used as organic solvents, the resulting organogels are called oleogels. In general, the solid lipids possess specific characteristics over liquid oils for food products. Thus far, high-melting lipids containing *trans* fatty acids, and saturated fatty acid moieties have been commonly employed. However, it has recently been claimed that the intake of high levels of *trans* and saturated fats contribute to global epidemics related to metabolic syndrome and cardiovascular disease [16]. The mechanistic resemblance of oleogels to *trans* and saturated fats makes liquid oil gelation an ideal alternative in developing fat-based food products. In practical application, both the mechanical and aesthetic properties of oleogels are vital for food and other products. Modulating various parameters for ideal mechanical and aesthetic properties has gained significant interest in recent years. Currently, a lot of research has been conducted on oleogels, to discover the relationship between gelator structure, the nature of the oil, and gelation ability [17,18]. However, it is still not well understood, due to the complicated systems. A better understanding of these oleogel systems will help to improve applications in cosmetic products, drug delivery and pharmaceuticals, and provide an alternative for making healthy foods that are free of trans fats and contain minimal saturated fats [19].

Among many small molecular gelators reported, amino-acid-based compounds have received tremendous attention because they are available in large quantities as inexpensive starting materials, and synthetic methods are relatively simple and well-established [20,21,22]. The demand for better and environmentally friendly gelators makes amino-acid-based amphiphiles very attractive, as these compounds are generally biodegradable and biocompatible. There is at least one amide linkage between the hydrophobic tail and the polar headgroup in the structure of amino-acid-based gelators, providing one of the interactions responsible for their supramolecular assembly, i.e., hydrogen bonding [23,24]. The gelation ability also depends on the hydrophobic interactions (van der Waals force) [25,26]. Seeking a balance between hydrogen bonding and van der Waals forces is very important for gelation.

N^α^, N^ε^-diacyl-l-lysine and its derivatives are called all-powerful gelators [27,28,29]. Its ester derivatives can form organogels in a wide variety of organic fluids, such as alkanes, alcohols, ketones, esters, cyclic ethers, aromatic solvents, polar solvents, and mineral and vegetable oils. Its carboxylate derivatives can form hydrogels in an aqueous solution. N^α^, N^ε^-diacyl-l-lysine itself has also displayed good organogelation ability. In this study, four N^α^, N^ε^-diacyl-l-lysines (i.e., N^α^, N^ε^-dioctanoyl-l-lysine; N^α^, N^ε^-didecanoyl-l-lysine; N^α^, N^ε^-dilauroyl-l-lysine; and N^α^, N^ε^-dimyristoyl-l-lysine) were synthesized and used as gelators to prepare oleogels in four kinds of vegetable oils. As the hydrogen-bonding and hydrophobic interactions between gelators and vegetable oils are responsible for the gelation behaviors and the properties of formed oleogels, the chemical structures of both N^α^, N^ε^-diacyl-l-lysines (i.e., the carbon-chain length) and oils (i.e., the fatty acid composition) are the key parameters of oleogel systems. The rheological properties, morphology, and oil-binding capacity of formed oleogels were studied, with hopes of providing better insight into the gelation abilities of l-lysine-based gelators with different carbon-chain lengths in different oils.

## 2. Materials and Methods

### 2.1. Materials

Octanoyl chloride (99%), decanoyl chloride (98%), lauroyl chloride (98%), myristoyl chloride (98%), and l-lysine (98%) were purchased from Shanghai Macklin Biochemical Co., Ltd. (Shanghai, China). Petroleum ether (bp 60–90 °C), anhydrous ethanol (99.8%), sodium hydroxide (≥98%), and hydrochloric acid (37%) were obtained from Beijing InnoChem Science & Technology Co., Ltd. (Beijing, China). Corn germ oil was purchased from Shandong Xiwang Food Co., Ltd. (Tianjin, China). Soybean oil was purchased from COFCO Fulinmen Food Marketing Co., Ltd. (Zouping, China). Olive oil was purchased from Shanghai Jiage Food Co., Ltd. (Shanghai, China). Linseed oil was purchased from Hebei Jiafeng vegetable oil Co., Ltd. (Handan, China). All the reagents were used as received.

### 2.2. Synthesis and Characterization of N^α^, N^ε^-diacyl-l-lysines

The l-lysine and fatty acid chloride were weighed in a molar ratio of 1:2. l-lysine was added into a four-necked flask, followed by a mixture of anhydrous ethanol and deionized water (2:1, *v*/*v*). The reaction mixture was cooled in an ice-water bath under stirring. Then fatty acid chloride was added slowly via a dropping funnel. In the meantime, the pH of the reaction solution was adjusted by an aqueous sodium hydroxide solution (10 wt%) and maintained at 9–10. The reaction continued for 3 h after the dropwise addition. At last, the reaction solution was taken out and allowed to stand at room temperature for 3–4 h. After acidification with dilute hydrochloric acid, the precipitate was filtered and washed with deionized water, then rinsed with petroleum ether three times, giving a white, solid appearance the final product.

Four N^α^, N^ε^-diacyl-l-lysines were characterized by FT-IR, ESI-MS, and ^1^H NMR measurements. IR spectra were obtained on a Nicolet iS10 FT-IR Spectrometer (Thermo Fisher Scientific, Madison, WI, USA) using KBr tablets at room temperature. ESI-MS spectra were recorded using an API3200 triple-quadrupole mass spectrometer (AB SCIEX, Redwood, CA, USA). The ^1^H NMR spectra were acquired on a DRX-600 NMR spectrometer (Bruker, Ettlingen, German).

### 2.3. Oleogel Preparation

The oleogel formation of N^α^, N^ε^-diacyl-l-lysines was investigated by the heating–cooling method [30]. The gelator was added to vegetable oil in a sealed vial. Then, the mixture was heated until a transparent solution was obtained, then cooled to room temperature by simply removing it from the heat. The gelation properties were evaluated by inversion of the vial, and the absence of gravitational flow was utilized to determine the successful achievement of gelation. The minimum gel concentration (MGC) for a gelator was determined by weighing up the minimum amount of gelator needed to form a stable gel through the heating–cooling cycle.

### 2.4. Characterization of Vegetable Oils

Analysis of the fatty acid composition of the oils was carried out using a 7890A/5975C gas chromatography–mass spectrometry system (GC-MS, Agilent, Santa Clara, CA, USA) equipped with a HP-5MS (30 m × 0.25 mm × 0.25 µm) capillary column. Helium (99.999%) was the carrier gas, with a flow rate of 1.2 mL/min. The initial temperature of 100 °C was held for 2 min and increased to 220 °C at 10 °C/min. The temperature was held at 220 °C for 1 min and increased to 230 °C at 2 °C/min, then held 2 min. Oils were trans-esterified to fatty acid methyl esters for GC-MS analyses. The injection (1 µL) was performed in the split mode at a split ratio of 1:20. The samples were run by electron ionization, and the source temperature and electron energy were 230 °C and 70 eV. The auxiliary heater temperature was 200 °C.

### 2.5. Rheological Behavior Measurements

The rheological analysis of the oleogels was measured by HAAKE MARS III rheometer (Thermo Electron GmbH, Dreieich, Germany). The time dependences of the storage modulus (G′) and loss modulus (G″) of oleogels were measured at 25 °C with a fixed rotational speed of 1 rad/s and a shear strain of 0.25 Pa. The temperature dependence of the storage modulus (G′) and loss modulus (G”) for oleogels was determined at a heating temperature from 20 °C to 145 °C, with a constant rotational speed of 1 rad/s and a shear strain of 0.25 Pa.

### 2.6. TEM Measurements

The morphology of oleogels was observed using TEM measurements. The samples were prepared according to the literature method of [21]. Firstly, a small amount of oleogel was cast on carbon-coated copper grids (300 mesh), then the sample was vacuum dried at 50 °C, 0.08 MPa for 2 days to evaporate the oils before observation. The microscopic images of oleogels were monitored using an X-MAX JEM-2100 transmission electron microscope (JEOL, Tokyo, Japan), operating at 120 kV.

### 2.7. Oil-Binding Capacity (OBC) Determination

The oil-binding capacity was measured according to the literature method of [21,31]. This methodology allows us to understand the degree of oil retention in the oleogel structure. First, 2 mL of the melted oleogel (the concentration corresponded to the MGC of each gelator) was put into a previously weighed (m_1_) centrifuge tube and kept at room temperature for 24 h. Then, the tube was weighed (m_2_) again. Finally, the tube was centrifuged at 10000 rpm for 15 min at room temperature and turned over to drain the released liquid oil. After drainage, the tubes were weighed (m_3_) again. The oil-binding capacity (OBC) values were calculated by the following equations.
(1)$$\mathrm{Released}\;\mathrm{oil}\;\left(\%\right)=\frac{{(m}_{2}{-m}_{1}{)-(m}_{3}{-m}_{1})}{m_{2}{-m}_{1}}\times 100$$
(2)OBC (%)=100−Released oil (%)

## 3. Results and Discussion

### 3.1. Preparation and Characterization of l-Lysine-Based Gelators

Four N^α^, N^ε^-diacyl lysine-based gelators were synthesized using the Schotten–Baumann reaction previously reported in the literature with minor modifications [32,33]. Briefly, fatty acyl chlorides were used as acyl donors, and reacted with l-lysine under alkaline conditions, followed by acidification with HCl aqueous solution. Four fatty acyl chlorides, i.e., octanoyl chloride, decanoyl chloride, lauroyl chloride, and myristoyl chloride, were applied in this paper. Both the α amino group and the ε amino group of l-lysine readily reacted with a fatty acyl chloride, forming chemically stable amide bonds. Four N^α^, N^ε^-diacyl l-lysine-based gelators were obtained with a yield of 42%−71% (Table 1). In the case of N^α^, N^ε^-dimyristoyl-l-lysine, the reason for the low yield was perhaps a low solubility of the reactants in the solvent used, due to the longer carbon-chain length.

The synthesized four N^α^, N^ε^-diacyl lysines were characterized by FT-IR, ESI-MS, and ^1^H NMR. The results are presented below.

N^α^, N^ε^-dioctanoyl-l-lysine (C8-lys-C8). IR (*v*_max_, cm^−1^): 3326.75 (N-H), 1724.29 (C=O), 1639.58 (C=O, amide band I), 1536.28 (δN-H, amide band II). ESI-MS: *m/z* = 396.8, which was assigned as [M − H]^−^, while M is molar weight of N^α^-octanoyl-N^ε^-octanoyl-l-lysine. ^1^H NMR (DMSO-*d*_6_, 600 MHz) δppm: 0.85 (t, *J* = 6.6 Hz, 6H, 2CH_3_), 1.23 (m, 18H, 9CH_2_), 1.35 (m, 2H, CH_2_), 1.47 (m, 2H, CH_2_), 1.54 (m, 2H, CH_2_), 1.66 (m, 2H, CH_2_), 2.02 (t, *J* = 7.8 Hz, 2H, CH_2_), 2.10(t, *J* = 6.0 Hz, 2H, CH_2_), 2.99 (m, 2H, CH_2_), 4.12 (m, 1H, CH), 7.72 (t, *J* = 4.8 Hz, 1H, NH), 7.97 (d, *J* = 7.8 Hz, 1H, NH), 12.41 (s, 1H, COOH).

N^α^, N^ε^-didecanoyl-l-lysine (C10-lys-C10). IR (*v*_max_, cm^−1^): 3326.75 (N-H), 1724.29 (C=O), 1639.58 (C=O, amide band I), 1536.28 (δN-H, amide band II). ESI-MS: *m/z* = 396.8, which was assigned as [M − H]^−^, while M is molar weight of N^α^-octanoyl-N^ε^-octanoyl-l-lysine. ^1^H NMR (DMSO-*d*_6_, 600 MHz) δppm: 0.85 (t, *J* = 6.6 Hz, 6H, 2CH_3_), 1.23 (m, 26H, 13CH_2_), 1.35 (m, 2H, CH_2_), 1.46 (m, 2H, CH_2_), 1.54 (m, 2H, CH_2_), 1.66 (m, 2H, CH_2_), 2.01 (t, *J* = 7.2 Hz, 2H, CH_2_), 2.10 (t, *J* = 7.2 Hz, 2H, CH_2_), 2.99 (m, 2H, CH_2_), 4.12 (m, 1H, CH), 7.72 (t, *J* = 5.4 Hz, 1H, NH), 7.97 (d, *J* = 7.8 Hz, 1H, NH), 12.44 (s, 1H, COOH).

N^α^, N^ε^-dilauroyl-l-lysine (C12-lys-C12). IR (*v*_max_, cm^−1^): 3326.75 (N-H), 1724.29 (C=O), 1639.58 (C=O, amide band I), 1536.28 (δN-H, amide band II). ESI-MS: *m/z* = 396.8, which was assigned as [M − H]^−^, while M is molar weight of N^α^-lauroyl-N^ε^-lauroyl-l-lysine. ^1^H NMR (DMSO-*d*_6_, 600 MHz) δppm: 0.85(t, *J* = 7.2 Hz, 6H, 2CH_3_), 1.23 (m, 34H, 17CH_2_), 1.35 (m, 2H, CH_2_), 1.46 (m, 2H, CH_2_), 1.54 (m, 2H, CH_2_), 1.65 (m, 2H, CH_2_), 2.01 (t, *J* = 7.2 Hz, 2H, CH_2_), 2.10 (t, *J* = 6.0 Hz, 2H, CH_2_), 2.99 (m, 2H, CH_2_), 4.12 (m, 1H, CH), 7.71 (t, *J* = 5.4 Hz, 1H, NH), 7.96 (d, *J* = 7.8 Hz, 1H, NH), 12.41 (s, 1H, COOH).

N^α^, N^ε^-dimyristoyl-l-lysine (C14-lys-C14). IR (*v*_max_, cm^−1^): 3326.75 (N-H), 1724.29 (C=O), 1639.58 (C=O, amide band I), 1536.28 (δN-H, amide band II). ESI-MS: *m/z* = 396.8, which was assigned as [M − H]^−^, while M is molar weight of N^α^-myristoyl-N^ε^-myristoyl-l-lysine. ^1^H NMR (DMSO-*d*_6_, 600 MHz) δppm: 0.85 (t, *J* = 6.6 Hz, 6H, 2CH_3_), 1.23 (m, 42H, 21CH_2_), 1.35 (m, 2H, CH_2_), 1.46 (m, 2H, CH_2_), 1.54 (m, 2H, CH_2_), 1.66 (m, 2H, CH_2_), 2.01 (t, *J* = 7.2 Hz, 2H, CH_2_), 2.09 (t, *J* = 6.6 Hz, 2H, CH_2_), 2.99 (m, 2H, CH_2_), 4.12 (m, 1H, CH), 7.71 (t, *J* = 4.8 Hz, 1H, NH), 7.97 (d, *J* = 7.8 Hz, 1H, NH), 12.45 (s, 1H, COOH).

### 3.2. Preparation of Oleogels

The four N^α^, N^ε^-diacyl-l-lysines with different carbon chains can form oleogels in four vegetable oils, i.e., linseed oil, soybean oil gel, corn germ oil, and olive oil, and the gel pictures are shown in Figure 1. All the oleogels did not flow after the vial was inverted, which means N^α^, N^ε^-diacyl-l-lysines can be used as effective gelators for vegetable oil structuring. As in Figure 1, gels formed in soybean oil and corn germ oil were white or translucent, while light yellow or yellow gels were obtained in linseed oil and olive oil. A similar phenomenon was reported in the literature [21], which may be related to the color of vegetable oils.

The MGC values of all the above-mentioned oleogel samples are listed in Table 2. As can be seen from the data, the gelation ability of gelators is closely related to their chemical structure, i.e., the acyl carbon-chain length. In the same vegetable oil, MGC increased with the increase in the acyl carbon-chain length of the gelators. Stable gel can be formed by 1.08 wt% of C8-lys-C8 in olive oil, while the MGC of C14-lys-C14 in linseed oil was even higher than 10 wt%. On the one hand, this may be due to two long carbon chains in the gelator molecules. The greater steric hindrance of longer chains is not conducive to the formation of gels. On the other hand, the reason linseed oil showed the higher MGC may be due the composition of the fatty acids. This oil is mainly composed of linolenic acid (Table 3).

The gelation ability of the gelators was also affected by the type of vegetable oil. For the same gelator, the MGC increased in the order of olive oil, corn germ oil, soybean oil, and linseed oil, indicating the strongest gelation ability in olive oil. This was caused by the different compositions of the four vegetable oils. It can be seen from Table 3 that the fatty acids of the four vegetable oils are mainly composed of saturated fatty acids and unsaturated fatty acids, of which unsaturated fatty acids account for the largest proportion [21]. Unsaturated fatty acids include oleic acid, linoleic acid, and linolenic acid, which contain one double bond, two double bonds, and three double bonds, respectively. Combined with MGC values in Table 2, there is a certain positive correlation between the gelling ability and the percentage of oleic acid. However, the compositon of vegetable oils was really complicatied. The differences in the degree of unsaturation and carbon-chain length, as well as different proportions, affected the intermolecular hydrogen bonding and hydrophobic interaction between the vegetable oils and gelators, leading to different gelation abilities of the gelators in different vegetable oils.

### 3.3. Rheological Properties

The viscoelastic characteristics of the oleogels formed by N^α^, N^ε^-diacyl-l-lysines (at the MGC mass fraction) were investigated by rheological measurements. The results of storage modulus (G′) and loss modulus (G″) are presented in Figure 2. In each case, the storage modulus (G′) is larger than the loss modulus (G”), and the storage modulus G′ is mainly independent on time, thus indicating the formation of stable oleogels.

When C8-lys-C8 was used as a gelator, a slightly higher G′ value of linseed oil gel and soybean oil gel, compared with corn germ oil gel and olive oil gel, indicates the higher mechanical strength due to the different composition of four vegetable oils. When C10-lys-C10 was used as a gelator, an obviously higher G′ value of linseed oil gel was observed among the four oleogels. Then comes corn germ oil gel, followed by soybean oil gel. Olive oil gel presented the lowest G′ value, indicating the weakest gel strength. As for C12-lys-C12, the G′ values of soybean oil gel and linseed oil gel were much higher than the other two oleogels, even ten times that of oleogel formed in corn germ oil. Similarly, when C14-lys-C14 acted as a gelator, soybean oil gel possessed the strongest mechanical strength due to the highest G′ value. However, the G′ value of linseed oil gel was as low as that of corn germ oil gel.

The mechanical strength of oleogels did not show apparent regularity with the vibration of the carbon-chain length of gelators and the type of vegetable oils. This is likely because of the complicated composition of the vegetable oils. Moreover, the amount of gelator in each gel sample was different, as there was a different MGC for each gelator.

The thermal stability of the oleogels was then evaluated. The mechanical properties of the oleogels as a function of temperature are presented in Figure 3. The rheological measurements indicated that the gels formed in vegetable oils at an MGC of C8-lys-C8 were stable up to 100 °C; above this temperature, the supramolecular interactions responsible for the formation of the gel weaken, and the sample melts to a viscous liquid. Similarly, the gels formed in corn germ oil and linseed oil at an MGC of C10-lys-C10 presented high thermal stability, keeping the gel-like consistency with mainly stable G′ and G″ values until about 130 °C and 100 °C. However, the gels formed in olive oil and soybean oil at an MGC of C10-lys-C10 displayed poor thermal stability, with slowly declining G′ values from about 30 °C, which then drastically decreased after 100 °C. As for C12-lys-C12 and C14-lys-C14, all the gels formed in vegetable oils were much more unstable, with a rapid decrease in G′ and G″ values with the increase in temperature.

For each gel, the values of G′ and G″ gradually decreased when the temperature increased. G′ was higher than G″ at the beginning, indicating that its gel state with solid behavior dominates. With the temperature continuing to rise, G″ surpassed G′, with the gel liquid behavior dominating. The intersection of the G′ and G″ curve can be defined as a phase transition point. The temperature at this point was summarized in Table 4. The phase transition temperatures of oleogels formed by C8-lys-C8 are all higher than 100 °C, while those of C12-lys-C12 and C14-lys-C14 gels were in the range of 40–75 °C, showing that the thermal stability of oleogels are mainly related to the carbon-chain length of gelators.

### 3.4. Morphology of Oleogels

In order to observe the network structure of oleogels produced with different gelators and vegetable oils, TEM was utilized, and the images are shown in Figure 4 and Figure 5. Long fibers were observed in the olive oil gel network when C8-lys-C8 was used as gelator at MGC (Figure 4A), while shorter fibers were displayed in olive oil gel with C10-lys-C10 at MGC (Figure 4B). However, olive oil gels formed by C12-lys-C12 and C14-lys-C14 yielded needle-like crystals (Figure 4C,D), which are a desirable feature for gel formation, as such crystals entrap a larger amount of liquid oil [16,34,35]. The needle-shaped crystals were about 2–3 µm long in the olive oil gel formed by C12-lys-C12, and about 1–2 µm long in the olive oil gel formed by C14-lys-C14.

Figure 5 presents the TEM images of the crystal network of oleogels formed by C14-lys-C14 in four vegetable oils, which show that all oleogel crystals were needle-like, regardless of oil type. For the four tested samples, the dimensions of the needle-like crystals were apparently unaffected by the oil type.

### 3.5. OBC of Oleogels

The OBC evaluation relates the strength of oleogels with the ability to retain the oil in the oleogel structure, showing the capacity of the oleogel structure to entrap oil. The OBC results of different vegetable oil gels are shown in Figure 6. We can see from Figure 6 that, at MGC concentration, the OBC of the oleogels is related to both the gelator and the vegetable oil type. When C8-lys-C8 was used as a gelator, the soybean oil gel showed the best capacity for entrapping oil, while the corn germ oil gel showed a weak oil-binding capacity. When C10-lys-C10 was used as a gelator, the linseed oil gel showed the weakest oil-entrapping capacity. As for C12-lys-C12, the order of oil-entrapping capacity in different vegetable oils was consistent with C8-lys-C8. While for C14-lys-C14, the order was consistent with C10-lys-C10. In general, C14-lys-C14 as a gelator performed the best in entrapping oils. This might be due to the sufficient amount of gelator in vegetable oils, because of the highest MGC for C14-lys-C14. Contrasting the rheological properties of gels in Figure 2, there is a positive correlation between the OBC values and the strength of the gels formed by C8-lys-C8, C12-lys-C12 and C14-lys-C14, but no such correlation was found for C10-lys-C10 gels.

Similarly, the OBC increased with increased gelator concentration within a certain range, and the ability to retain oil was enhanced [21]. For all four gelators, the highest OBC was obtained in soybean oil, followed by olive oil. This is likely owing to the different compositions of vegetable oils, which in turn, affects the interaction between gelators and vegetable oil molecules, resulting in the difference in OBC.

The oil-binding capacity of an oleogel is perhaps its most important measure of functionality since it will dictate which applications may benefit from oil gelation. For example, oleogels have been considered as fat replacements for meat products. An oleogel with a low oil-entrapping capacity would easily exude oil into the surrounding food matrix upon chewing, altering the textural and sensory properties of the product in an undesirable manner [36].

## 4. Conclusions

Four N^α^, N^ε^-diacyl-l-lysines (i.e., N^α^, N^ε^-dioctanoyl-l-lysine; N^α^, N^ε^-didecanoyl-l-lysine; N^α^, N^ε^-dilauroyl-l-lysine; and N^α^, N^ε^-dimyristoyl-l-lysine) were synthesized through the Schotten–Baumann reaction. The four compounds can be applied as organogelators for preparing oleogels in olive oil, corn germ oil, soybean oil, and linseed oil. The gelation ability is not only related to the structure of the gelator, but is also affected by the composition of the oil. The MGC increased with the increase in the acyl carbon-chain length of the gelators. For each gelator, the strongest gelation ability was displayed in olive oil. The rheological properties, morphology, and oil-binding capacity of the formed oleogels were also affected by both gelators and oil types, because the hydrophobicity and H-bonding provided by the alkyl chain and amide group are probably responsible for the gelation of N^α^, N^ε^-diacyl-l-lysines in vegetable oils.

## Figures and Tables

**Figure 1 molecules-27-01369-f001:**
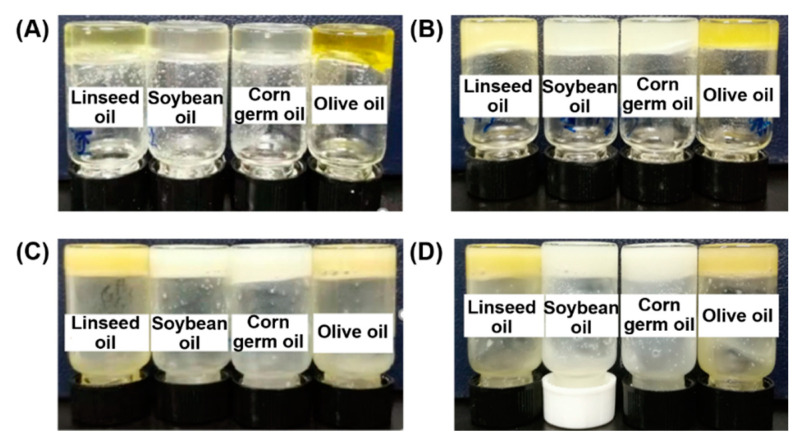
Oleogels formed by lysine-based gelators in linseed oil, soybean oil gel, corn germ oil and olive oil: (**A**) C8-lys-C8, (**B**) C10-lys-C10, (**C**) C12-lys-C12, (**D**) C14-lys-C14.

**Figure 2 molecules-27-01369-f002:**
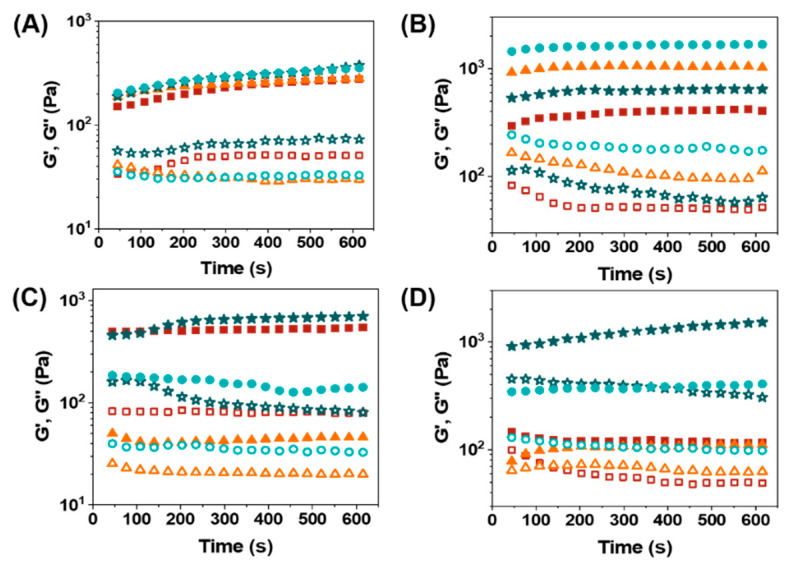
Variation of storage modulus (G′) and loss modulus (G″) for oleogels formed by l-lysine-based gelators: (**A**) C8-lys-C8, (**B**) C10-lys-C10, (**C**) C12-lys-C12, and (**D**) C14-lys-C14 at 25 °C. Solid squares—Olive oil gel (G′); Open squares—Olive oil gel (G″); Solid triangles—Corn germ oil gel (G′); Open triangles—Corn germ oil gel (G″); Solid stars—Soybean oil gel (G′); Open stars—Soybean oil gel (G″); Solid circles—Linseed oil gel (G′); Open circles—Linseed oil gel (G″).

**Figure 3 molecules-27-01369-f003:**
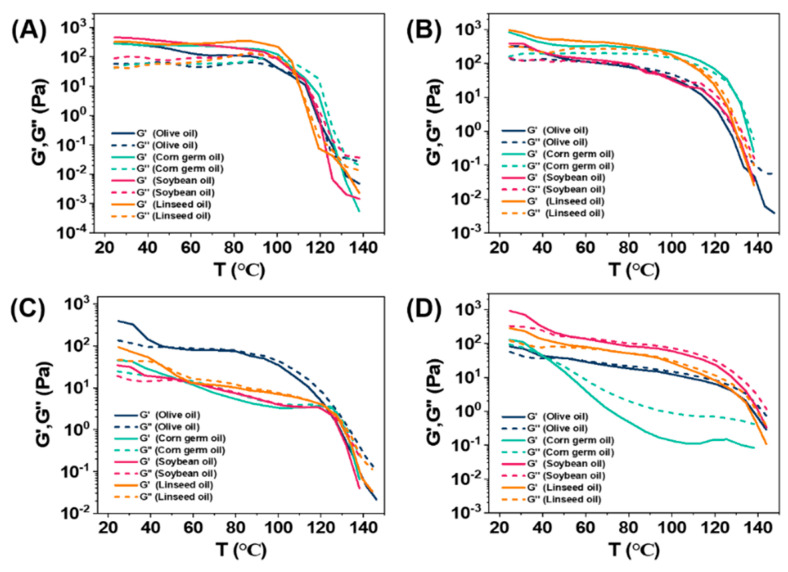
Dependence of the storage modulus (G′) and loss modulus (G″) with the temperature for a gelator mass fraction of corresponding MGC in four vegetable oils: (**A**) C8-lys-C8, (**B**) C10-lys-C10, (**C**) C12-lys-C12, and (**D**) C14-lys-C14.

**Figure 4 molecules-27-01369-f004:**
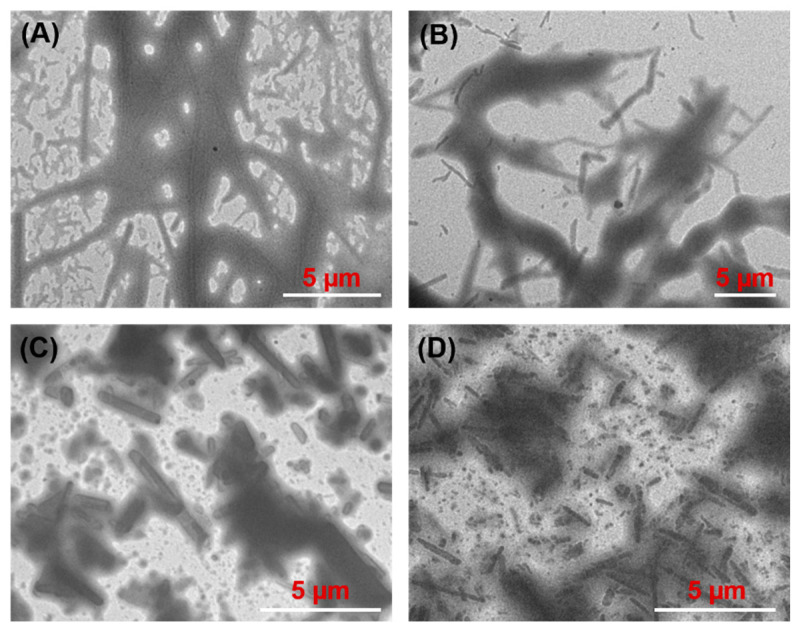
TEM images of oleogels prepared by lysine-based gelators in olive oil: C8-lys-C8 (**A**), C10-lys-C10 (**B**), C12-lys-C12 (**C**), and C14-lys-C14 (**D**).

**Figure 5 molecules-27-01369-f005:**
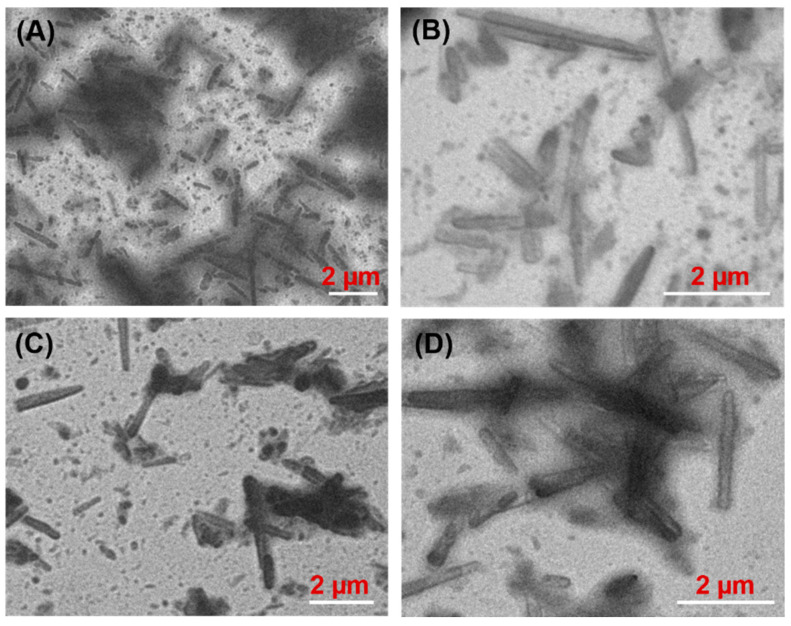
TEM images of oleogels prepared by C14-lys-C14 in olive oil (**A**), corn germ oil (**B**), soybean oil (**C**), and linseed oil (**D**).

**Figure 6 molecules-27-01369-f006:**
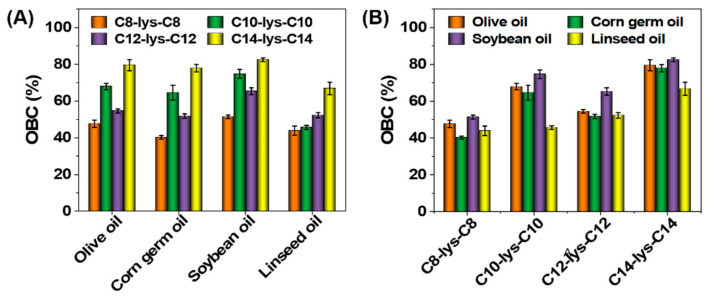
OBC of oleogels formed by four l-lysine-based gelators in different vegetable oils: (**A**) Comparison of different gelators, (**B**) Comparison of different oils.

**Table 1 molecules-27-01369-t001:** Product list of four N^α^, N^ε^-diacyl lysine-based gelators.

Gelator	Structure	Abbreviation	Yield (%)
N^α^, N^ε^-dioctanoyl-l-lysine	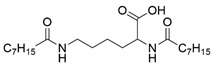	C8-lys-C8	69
N^α^, N^ε^-didecanoyl-l-lysine	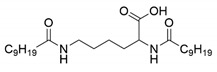	C10-lys-C10	71
N^α^, N^ε^-dilauroyl-l-lysine	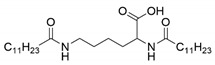	C12-lys-C12	65
N^α^, N^ε^-dimyristoyl-l-lysine	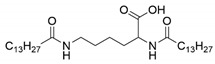	C14-lys-C14	42

**Table 2 molecules-27-01369-t002:** Gelation ability of four gelators in vegetable oils.

Vegetable Oils	MGC (wt%)			
	C8-lys-C8	C10-lys-C10	C12-lys-C12	C14-lys-C14
Olive oil	1.08	1.28	4.28	7.79
Corn germ oil	1.18	1.93	4.37	8.63
Soybean oil	1.36	2.14	4.42	8.84
Linseed oil	1.67	2.27	4.55	10.57

**Table 3 molecules-27-01369-t003:** Fatty acid composition of four vegetable oils (%; mean values ± SD, *n* = 3).

Vegetable Oils	SFA ^1^	Oleic Acid	Linoleic Acid	Linolenic Acid
Olive oil	18.66 ± 0.11	74.55 ± 0.06	6.07 ± 0.02	0.72 ± 0.10
Corn germ oil	17.01 ± 0.14	30.26 ± 0.03	51.70 ± 0.07	1.03 ± 0.21
Soybean oil	18.82 ± 0.21	26.83 ± 0.18	47.40 ± 0.08	6.95 ± 0.23
Linseed oil	14.04 ± 0.08	22.15 ± 0.30	14.90 ± 0.19	48.91 ± 0.03

^1^ SFA, saturated fatty acids, which mainly contained palmitic acid and stearic acid.

**Table 4 molecules-27-01369-t004:** The phase transition temperature of oleogels formed by l-lysine-based gelators.

Vegetable Oils	Phase Transition Temperature (°C)			
	C8-lys-C8	C10-lys-C10	C12-lys-C12	C14-lys-C14
Olive oil	101.1	62.3	48.4	51.5
Corn germ oil	102.5	130.9	51.5	41.1
Soybean oil	102.5	84.2	60.8	57.0
Linseed oil	112.9	104.4	44.9	71.4

## Data Availability

Not applicable.

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
