# Peer review of "l-Lysine-Based Gelators for the Formation of Oleogels in Four Vegetable Oils"

_molecules, 2022, doi:10.3390/molecules27041369_

Round 1

Reviewer 1 Report

Reviewing of: L-Lysine-based gelators for the formation of oleogels in four vegetable oils

The authors have synthetized and studied several modified Lysine amino acids for gelation experiments in vegetable oils. They have studied the influence of the carbon chain length on the gelation abilities of their low molecular gelators and they have described some physico-chemical characterisitics of the corresponding oleogels. The various experiments are well presented and the authors have drawn conclusions from their observations.

Yet, I found that this work is a bit too superficial and this study would have required some more physical characterizations. Indeed, the introduction does not state clearly the interest of oleogels in the food industry for example. As I found this topic original, I would have been interested in reading the current challenges that  the community is facing, specifically into the world of oleogels. As an example of the superficial aspect of this work, the conclusion states “The gelation ability is not only related to the structure of the gelator but also affected by the composition of the oil” (lines 301-303) which could have been anticipated from the beginning to me. More there is no detail about the composition of the oils used in this study and no variation of their composition was attempted to clearly demonstrate the influence of the oil's nature.

I also recommend the authors to perform other characterizations since the first goal of the article (“The objective of this study was to determine the effect of carbon chain length on the organogelation ability…” line 57) is not achieved to my opinion. The authors could have done some FTIR of the gel, to demonstrate what was the origin of the gelator assembly within the vegetable oils. Although it is less common and accessible, SAXS experiment would bring a lot of information, especially in terms of supramolecular packing. It seems they have obtained crystalline materials from the TEM picture, which may respond to SAXS experiments. Did the authors think about such an alternative?

Author Response

Dear Reviewer,

Thank you very much for reviewing our manuscript and for the constructive comments, which greatly helped us to improve the manuscript. We have read the comments carefully and have made revisions to the manuscript. All revisions to the manuscript have been marked up using the “Track Changes” function of MS Word, such that the changes can be viewed in the revised manuscript. And point-by-point responses are attached. We hope that the revisions and explanation are acceptable, and your favorable consideration of our manuscript is greatly appreciated.  Please see the attachment for details.

Best regards,

Yours sincerely,

Dr. Guiju Zhang

Reviewer 2 Report

The article describes the synthesis and characterization of four amino acid-based compounds (Nα,Nε-di-12 acyl-L-lysine derivatives, with chain lengths of 8,10,12 and 14), which were used as gelators of four different oils (olive, soybean, corn germ and linseed).

The authors also present a carefully characterization of the oleogels by measurement of the rheological behavior, TEM observation and determination of the oil binding capacity.

The clue reaction for the synthesis of the lysine derivatives was a Schotten-Baumann reaction using acyl chlorides.

The work is very interesting considering that oleogels is a novel and interest growing thematic. Compounds and gels obtained were fully characterized. Objectives and purpose of the research are clearly stablished.

However, I have several observations and questions.

  • In line 36, other new bibliography may be included. Recent citations will highlight the actuality of the topic. In a brief and quick search newer citations such as “Artificial cells, nanomedicine, and biotechnology 2020, 48, 1, 266–275” and “Materials Science & Engineering C 82 (2018) 80–90”, for instance, were found.
  • In line 78, when the compounds were synthesized, is it correct the molar relation between lysine and acyl chloride? If both amino groups were amidated, shouldn´t 2 equivalents of the chloride have been used?
  • In the case of Nα,Nε-dimyristoyl-L-lysine, which is the justification authors could give to the low yield. I think that perhaps a low solubility of the reactants in the solvent used may be the cause for this.
  • For all compounds optical rotation should be informed.
  • In line 186, the reason for why linseed oil showed the higher MGC may be due the composition of the fatty acids, as it is mention later. This oil is mainly composed by linolenic acid (18:3).
  • When rheological properties are described line 206 onwards (specially for C8-Lys-C8), which is the role of the concentration of the gelator in the properties? As the authors stated later the amount of gelator is different in each gel sample and it is sometimes difficult to state a tendence.
  • In table 3, is there any explanation for the variability observed for C10-Lys-C10. Perhaps no explanation could be given since it is a complicated system.

In general, this article is interesting, this reviewer does recommend minor revision in order to be published in Molecules.

Author Response

(The authors gave the same response as above.)

Reviewer 3 Report

This manuscript by Xu and co-workers describes the preparation and characterization of gels in vegetable oils, obtained form N-acyl-Lysine gelators. The work is well written, and the conclusions are sound. However, the work in my view should be complemented with further experiments, to be accepted for publication. Hence, major revision should be conducted, according to the following points.

1)  Page 1, line 32: it is not true that, besides oleogels, only hydro- and organogels are reported in the literature. Recently also ionic liquid gels i.e. gels obtained in ionic liquids (10.1039/C5GC02277K and 10.1039/C4SM01360C and 10.1016/j.jcis.2019.04.034) and eutectogels (gels obtained in deep eutectic solvents, 10.1016/j.gce.2021.06.001 and 10.1021/acssuschemeng.8b04278) have been described. Please add these in the introduction.

2) Section 2.5, line 114: what do the authors mean with “dried at room temperature”? Do they mean that oil is removed by evaporation? Is oil retained in the sample imaged? Please specify and clarify adding more detail on how the samples for TEM were prepared.

3) Section 2.6, line 120: I think there is a typo here, and “melted organogel” should be “melted oleogel”.

4) Section 3.3, rheology: I think that a mere comparison of G’ over 10 minutes at a fixed strain and frequency is not sufficient to evaluate the mechanical properties of the gels. Please run strain and frequency sweep measurements on the gels, so that the yield stress of the gels, i.e. the crossover point of G’ and G’’ can be obtained. It could be interesting to see if such values correlate or not with the oil binding capacity of the gel, given the fact the OBC is obtained after subjecting the gels to a considerable mechanical stimulus.

5) Why was OBC determined at the MGC and not at a common gelator concentration? Please motivate the reason for this choice.

6) Figure 6. Please report reproducibility or error bars in the OBC plots. Again, it would be interesting to see whether OBC values correlate or not with the strength of the gels.

Author Response

(The authors gave the same response as above.)

Reviewer 4 Report

In this study, four L-lysine derivative gelling agents have been synthesized and used in four types of vegetable oils. Both the synthesis and the gelation capacity of these have been well characterized. However, the four types of oils used have not been characterized, being able to shed light on the different interactions present and therefore on the gelation capacity.

For all these reasons, I believe it is convenient to characterize vegetable oils and study the correlation of their composition with their gelation ability.

Author Response

(The authors gave the same response as above.)

Round 2

Reviewer 1 Report

The revised manuscript contains more information in the introduction section to support the originality of this work.

The authors have tried to answer all the questions and remarks I had. I recommend to publish it as it is with a minor correction.

There is a tiny mistake,  line 318 Figyre must be replaced by Figure.

Reviewer 3 Report

The authors have provided suitable answers to the issues raised by the previous version. The manuscript can be accepeted in Molecules